# A Novel, Modular Robot for Educational Robotics Developed Using Action Research Evaluated on Technology Acceptance Model

**Avraam Chatzopoulos** [1,*] , **Michail Kalogiannakis** [2] , **Stamatis Papadakis** [2,*] **and Michail Papoutsidakis** [1]

1  Department of Industrial Design & Production Engineering, University of West Attica, 12243 Egaleo, Greece; mipapou@uniwa.gr
2  Department of Preschool Education, University of Crete, 74100 Rethymnon, Greece; mkalogian@uoc.gr
*  Correspondence: xatzopoulos@uniwa.gr (A.C.); stpapadakis@uoc.gr (S.P.)

**Abstract:** This research evaluates a novel, modular, open-source, and low-cost educational robotic platform in Educational Robotics and STEM Education. It is the sequel of an action research cycle on which the development of this robot is based. The impetus for the need to develop this came from the evaluation of qualitative and quantitative research data collected during an educational robotics event with significant participation of students in Athens, which showed an intense interest in students in participating in educational robotics activities, but—at the same time—recorded their low involvement due to the high cost of educational robots and robotic platforms. Based on the research's findings, this robot was designed to suit the whole educational community; its specifications came from its members' needs and the processing and analysis of qualitative and quantitative data. This paper presents an evaluation of the robot using the Technology Acceptance Model. The robot was exposed to 116 undergraduate students attending a pedagogical university department to evaluate its handling according to the model's factors. Research results were promising and showed a high degree of acceptance of the robot by these students and future teachers, providing the impetus for further research.

**Keywords:** educational robotics; STEM; open platforms; TAM

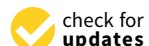



## 1. Introduction

Educational Robotics (ER) is a new trend introduced in education that enriches the learning environment and promotes knowledge [1]. ER creates a playful and helpful environment that increases students' interest in Science, Technology, Engineering, and Mathematics (STEM) activities and programming [2,3]. ER is an approach to STEM [4]. It refers to a broad collection of technology (robotic) platforms, educational activities, programs, resources, and learning theories [4,5]. ER offers a practical, hands-on understanding of the things we use in daily life and do not understand, such as motors, speakers, buzzers, lights, sensors (temperature, proximity, motion, and light), and problems related to hardware and software bugs [6]. ER methodology is based on designing and programming a robot by creating its parts. ER is believed to help students develop communication, social, mathematical, and motor skills and learn to code and program [6,7]. Besides, ER is used as an innovative teaching and learning tool that benefits students [2] to:

- understand objects and create multiple representations of them, increase their abstract design conception.
- develop students' high-level skills, boost their knowledge (by solving authentic problems), and improve their learning (via experimentation and research), especially in STEM; and
- to increase students' collaboration and develop their communication.

## 2. Educational Robot Platforms

The first move towards ER [4] is selecting and using an educational robot or robotic platform from the market with many buying options [8]. The choice of the proper ER robotic platform depends on the educational activities and fulfils students' specific educational aims, divided into two basic categories: (i) activities that build a robot, and (ii) activities to manage a robot. As a result, the ER robotic tools may be distinguished into two basic categories [1]:

1.  Programmable robots (e.g., Beebot, Edison, Ozobot, Thymio, mBot, and Blue-Bot) are usually mobile wheeled robots based on the original Papert's turtle and his Logo programming language concept [9]; and
2.  robotics construction kits, e.g., Lego® WeDo (Lego A/S, Billund, Denmark) and Mindstorms (Lego A/S, Billund, Denmark), Makeblock (Makeblock, Shenzhen, China), Robotis Dream Kits (Robotis INC, Lake Forest, CA, United States of America), VEX V5 (Innovation First International, Greenville, TX, United States of America), Hexbug VEX (Innovation First International, Greenville, TX, United States of America), Edison (Microbric, Grange, Australia), Thymio (Mobsya, Renens, Switzerland), etc. that are building blocks used to make a robot.

Besides, several robotic kits work and are programmed by popular microcontrollers, such as the BBC's Micro: Bit—a programmable device introduced for purely educational purposes [10]—the Arduino's boards [11–13], and the Raspberry Pi—a cheap computer, with a credit-card size, used to learn to program and to practice with projects. Undoubtedly, the above robots list is too limited, and there are plenty of other commercial products, such as MouseBot, KidFirstCoding, Evo, Tinkerbots, KUBO, and Pro-Bot, to name some, but they usually share some common limitations:

*   they are costly; thus, they are not affordable to all.
*   most of them cannot be expanded with more actuators and sensors.
*   they usually use proprietary-source (not open-source) for their software and hardware, so it is not easy to expand them by their community or third parties; and
*   some of them need an internet connection for their programming or use specific software (need for a program installation) or hardware (demand current hardware specifications).

## 3. The Necessity for a New Educational Robot Development

In 2019, researchers from the University of West Attica, in collaboration with the Municipality of Agia Varvara, handled a successful ER event to present STEM and ER Education to the local education community (teachers, parents, students, and stakeholders). Along with a presentation and an ER workshop, a survey was conducted to measure their interest in ER [14,15]. Pre- and post-questionnaires were distributed to record participants' views and to measure their interest in ER, and the results were fascinating [16]:

1.  most participants knew about educational robots (84%)—especially Lego (68%)—but they were unsure what ER was.
2.  Most of the participants (88%) would like to develop their educational robot, and the vast majority of the participants' parents (67%) would like to get involved in educational robots' development.
3.  The participants shaped the features of the "ideal educational robot": they wanted to program it using any device (54%), and compatible with older devices (40%), they wanted to be open-source in terms of software and hardware (92%), and half of them prefer to 3D-print it (50%); and
4.  Concerning the robot's cost, most of the participants (75%) were divided into two categories: 42.9% wanted the robot's cost to be under 50€, and 32.1% preferred it to be between 51€ and 100€.

Considering the above limitations of existing commercial educational robots and the needs of the educational community that emerged from the survey, the researchers in this article decided to design and build an educational robot based on the community's

suggestions. For this reason, action research was used as the most proper method to derive the characteristics that this robot should have.

## 4. Action Research

In the 1940s, Kurt Lewin and John Collier [17] introduced Action Research (AR); the research aims to involve researchers with social groups in making decisions on problems. In the early 1970s, AR's practices were applied to educational research to motivate teachers' professionalization improvement and transform them into researchers, aiming to improve their teaching practices [18]. AR can be used as a methodology for teachers to understand educational practices, generate knowledge, and examine their practice if needed to improve it [19]. In modern times, AR is used in many other scientific areas, including information systems [18], because it is mainly a short-range action carried out by the participants themselves and other people in the same community, aiming to practice [20]. Several AR models are available; almost all use the spiral or circular process [21,22]. The most common AR model introduces a spiral of cycles where each research stage includes the following phases: (i) planning, (ii) action, (iii) observation, and (iv) reflecting/evaluation, leading into further cycles where these stages are repeated [23–25]. Based on the above AR model, this research adapts it to design and develop an educational robot that is conducted by the participants themselves (teachers and students) and other stakeholders in the same community (parents and researchers) through the following phases in a self-reflective spiral of cycles (Figure 1):

1.  data collection and figure out the robot's specifications (planning stage).
2.  robot's design and development (action stage).
3.  robot's application on STEM and Educational Robotics (observation stage); and
4.  evaluation, reflection, and improvement suggestions to lead to another AR cycle (reflecting stage).

**Figure 1.** Application of Action Research in the robot development: AR's phases and cycles.

In this paper, the evaluation phase of the robot is presented so that the research can proceed to a new AR cycle.

## 5. The Proposed Educational Robot

The proposed educational robot shaped its initial specifications by the first cycle of an AR and the survey's data collection [26,27]. Its design focus on expandability, rapid development, and ease of use, so the following specifications had to be considered:

1. robots' costs must be low enough to be affordable by the community majority, especially not more than 50.00€.
2. the development of the robot should not hold difficult ("exotic") electronic parts and should be easy to assemble.
3. the robot will expand with future actuators and sensors and should be open-source.
4. Programming the robot should be easy enough: no need for an internet connection and software/app to download/install, and it will be programmed by most devices (smartphones, tablets, and PC). An embedded block-based language will be preferred as the tool for robot programming; and
5. the robot should be easily customized by the educational community, such as by designing new robot shells; thus, students to strengthen their imagination and enhance their artistic inclinations.

*5.1. Robot's Architecture*

The architecture block diagram of the robot is shown in Figure 2. The robot's operation is based on the principal part of the block diagram: the microcontroller, which manages many tasks:

- to provide the robot's Wireless Access Point (WAP with a unique SSID).
- to act as an intermediate web server serving the clients' (user's devices, e.g., PC, tablet, and smartphone) requests.
- to provide the robot's user interface (UI) to the users' devices.
- to read users' Visual Programming Language (VPL) commands and convert them to robot directions; and, finally,
- to incorporate all the necessary functions for smooth cooperation of the above operations.

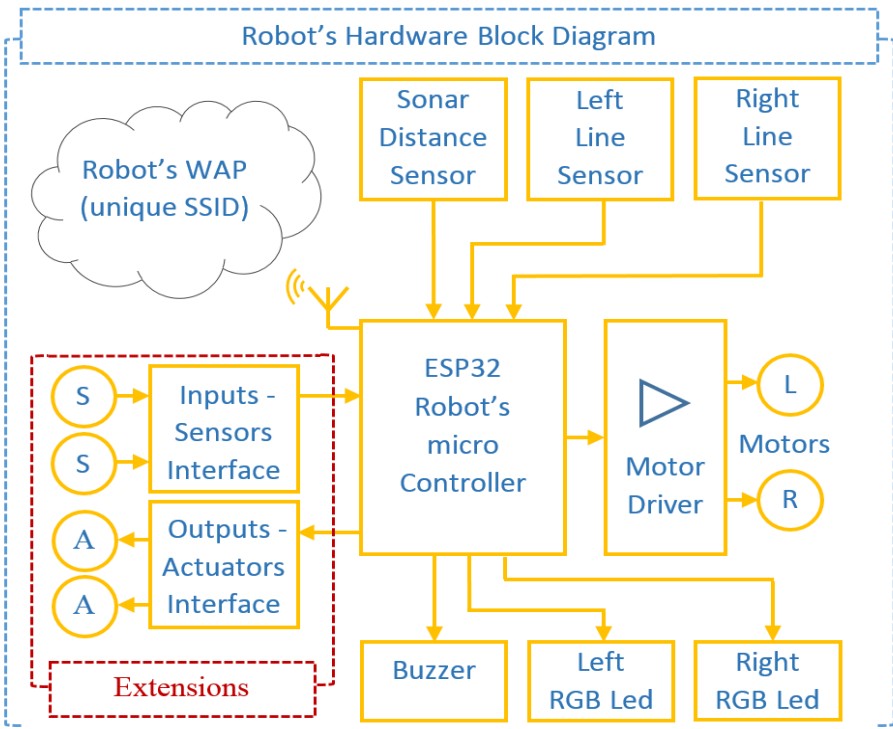

**Figure 2.** Robot's architecture block diagram.

It uses Espessif System's ESP32—a successor to the ESP8266. ESP32 was selected as the specific microcontroller is cheap, low-power, with integrated Bluetooth and Wi-Fi, and it can perform as a complete stand-alone system. It is equipped with a dual-core 32-bit microprocessor at 160 MHz, 448 KB ROM, 320 KB RAM, and includes 34 GPIOs, 12-bit ADCs, 8-bit DACs, touch sensors, SPI, I²S, I²C, UART, CAN, Infrared remote controller,

PWM, and an SD host controller, among others. In addition, it can be programmed in C/C++ through the Arduino IDE [12]. Although it was the primary idea, the famous and cheap Arduino UNO was not used because ESP32 is a better choice at a low cost and comes with enhanced features and peripherals. ESP32: (i) is connected to a motor driver to control the robot's motors [11]; (ii) is connected to the robot's actuators (LEDs, buttons, and buzzer) through interfaces to control them; and (iii) is connected to the robot's sensors (supersonic sonar, and line sensors) to read their signals. ESP32 operates efficiently under multi-tasking software to provide the robot's multi-role tasks.

In Figure 3, the robot's multi-user operation block diagram shows that multiple users (or clients in the corresponding terminology) can access and control a single robot. This multi-users operation is chosen because in many classrooms, the ratio of one robot available for every one or two student/s does not apply, and it is common for a single robot to be available for the whole class. So, the proposed robot supplies two operating options: (i) single-user operation where one user access one robot; or (ii) multi-user operation where many users access one robot in case there is only one robot available to the students' class, thus giving access to all students.

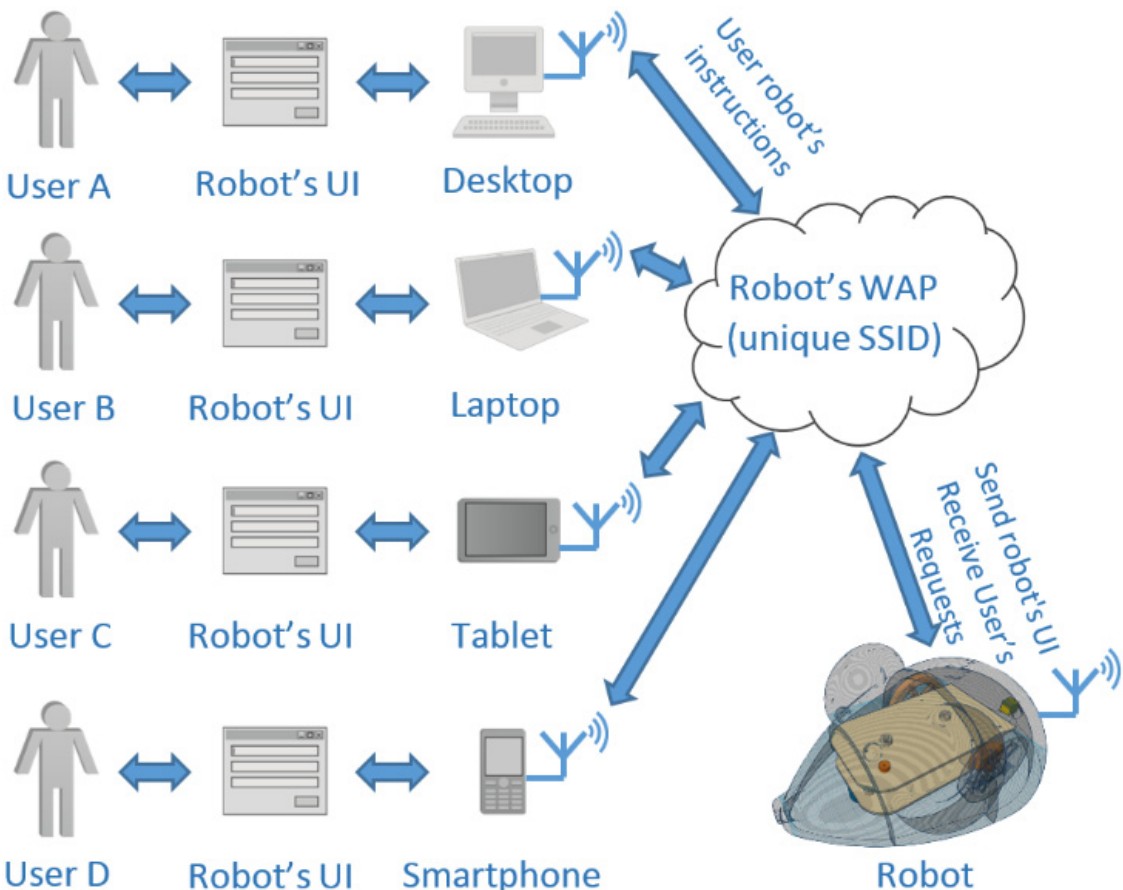

**Figure 3.** Robot's multi-users operation block diagram.

### 5.2. Robot Hardware

The robot is based on a 3D-printed chassis (Figure 4). Every part of the robot is 3D-designed and 3D-printed; thus, it is easy to be customized by the educational community. Its hardware consists of the 3D parts and other parts, such as two servo motors (converted to DC motors), a battery and a battery holder, various electronic parts, and other hardware for assembly. A complete list of the robot's components is shown in Table 1.

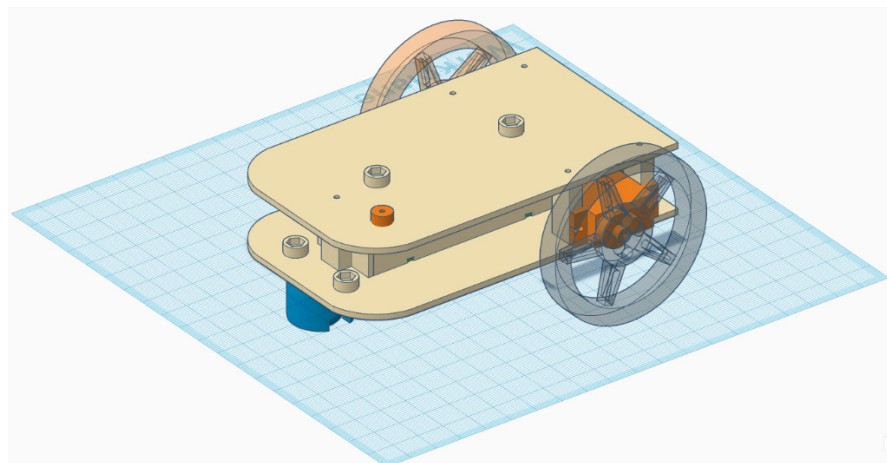

**Figure 4. The** robot's internal chassis design is shown where its external shell (shape and design) is totally absent.

**Table 1.** Robot's components list.

| Qty | Part's Name | Qty | Part's Name |
|---|---|---|---|
| 1 | Robot's Base (3D-printed with PLA) | 1 | Robot's Shell (3D-printed with PLA). It can be easily customized using Tinkercad© or other 3D software |
| 2 | Wheel with tire (3D-printed with PLA) | 1 | Ball caster wheel |
| 1 | ESP32 microcontroller | 1 | L293D motor driver IC |
| 2 | Micro servo motors (e.g., SG90 converted to DC) | 1 | Basic electronics board |
| 2 | TCRT5000 optical sensor (line sensor) | 1 | Sonar HC-SR04 (for distance meter) |
| 2 | RGB Led common cathode | 1 | Buzzer |
| 1 | TP4056 Li-Ion charger | 1 | 18650 3.6V 3350mAh Li-Ion battery |
| 13 | Resistors ($1 \times 15\ \Omega$, $4 \times 33\ \Omega$, $2 \times 56\ \Omega$, $2 \times 68\ \Omega$, $1 \times 470\ \Omega$, $1 \times 1\ K\Omega$, $2 \times 4.7\ K\Omega$) | 1 | Breadboard (to connect experimental electronic circuits) |
| 1 | USB cable (1x type B to micro-USB) | 40 | Jumper Wires (male/male, female/female) |

*5.3. Robot's Design*

The robot's shell is the housing that hides all the robot's electronic parts, giving it a more pleasant appearance. One of the robot's basic specifications was to be easily customized by the educational community in terms of design. Thus, students will be able to design the "ideal" for the robots by strengthening their imaginations and enhancing their artistic inclinations. So, its shell was designed using Autodesk's Tinkercad© (Autodesk, Inc, Mill Valley, California, United States), an online cloud-based, accessible, collaborative, 3D software that can be easily 3D-printed by a 3D printer. Many versions of the robot's shell were designed and evaluated, and some were freely 3D designs under the "Attribution CC BY" license [2,16].

In Figure 5, a perspective view of a mouse robot's shell, based on a 3D open design [28], is customized and transformed to host the robot's electronics. In Figure 6, the mouse robot shell reveals its hardware parts (mechanical parts and electronic circuits).

In Figure 7, some of the other robot experimental shells (based on animals or other figures).

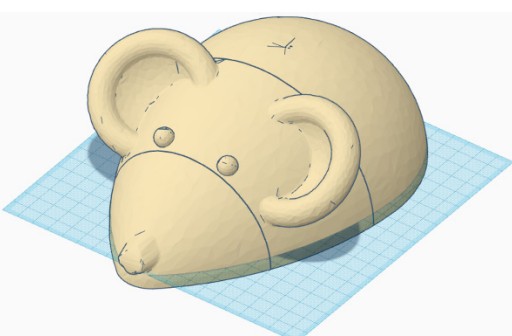

**Figure 5. The** robot's perspective view with the external "mouse" shell.

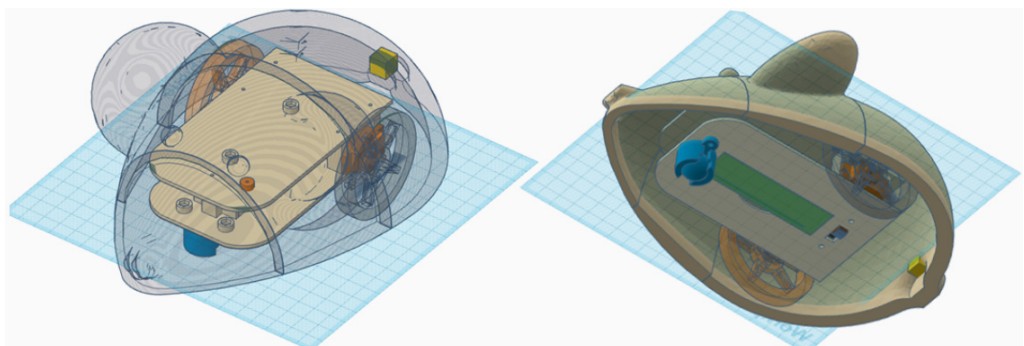

**Figure 6.** The image on the left shows the transparent view of the robot where the transparent "mouse" shell and its base are shown, while, on the right image is shown the bottom view of the robot where its wheels and battery can be seen.

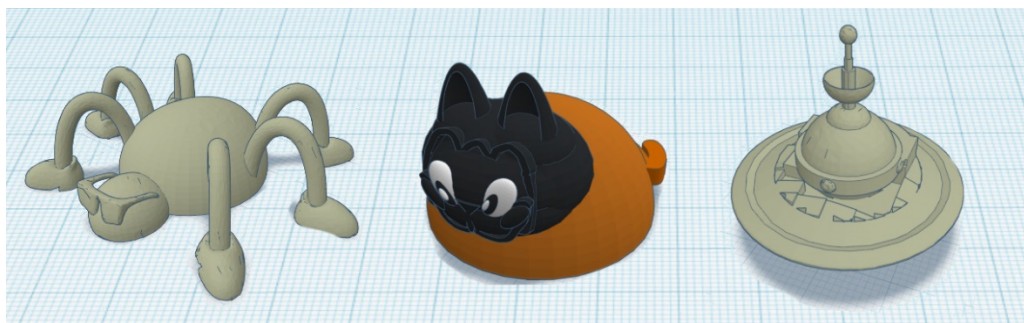

**Figure 7.** Experimental robot shells.

*5.4. Robot Software*

The robot's software consists of two parts: (i) its firmware that manages all its operations, and (ii) it is a programming language through which users can program it without having earlier programming experience. Some of the primary responsibilities of the robot's firmware are:

- to set up a Wireless Access Point (WAP).
- to host a web server that serves user's requests.
- to drive the robot's motors and controls its actuators.
- to read the data from the robot's sensors; and
- to work with all the above for a smooth operation.

The robot's programming language is an integrated custom visual programming language (VPL) based on Dethe Elza's Block code with a small memory footprint—the main advantage of choosing it [16,29]. It is simple, easy to learn, and integrated into ESP32, leaving enough memory for the user's programs and the robot's operations. Another advantage of using this VPL is the ability to be programmed using only the user's device

(smartphone, tablet, and PC) browser without the need to download/install any software or application because it is entirely written in HTML, Cascade Style Sheets (CSS), and

JavaScript. The robot's VPL supports three operational modes: Easy, Medium, and Hard.

Easy mode is to get to know and control the robot's movement by pressing the appropriate buttons. The user becomes familiar with the robot software's interface (buttons and icons) and may have fun with the robot and understand its movements (Figure 8).

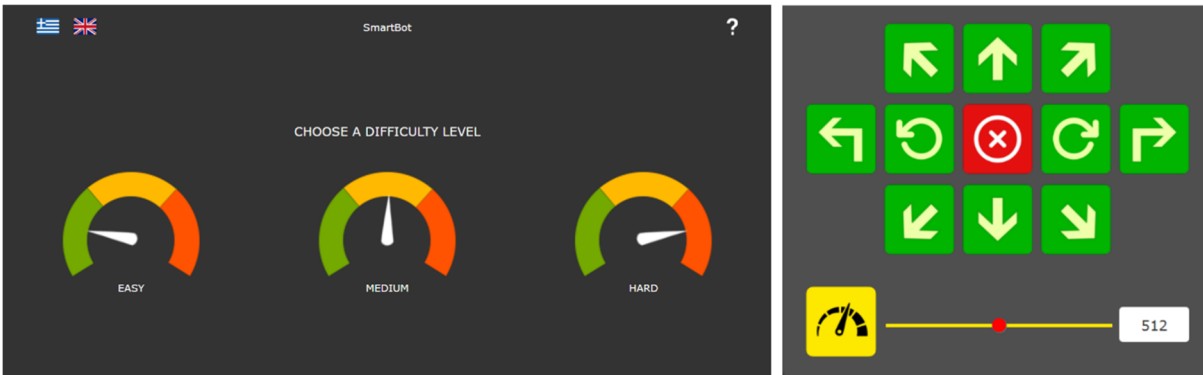

**Figure 8.** Robot's intro screen (**left**) and easy operation mode (**right**).

Medium mode is to program the robot through its VPL. As shown in Figure 9, the user edits the robot's program according to his needs using the VPL's blocks, which are non-text representations that can be dragged around the screen, attached to others, and chained together, being the robot's code to be executed. The program starts when the user presses the Start Block, and it stops when the running sequence executes a Stop Block.

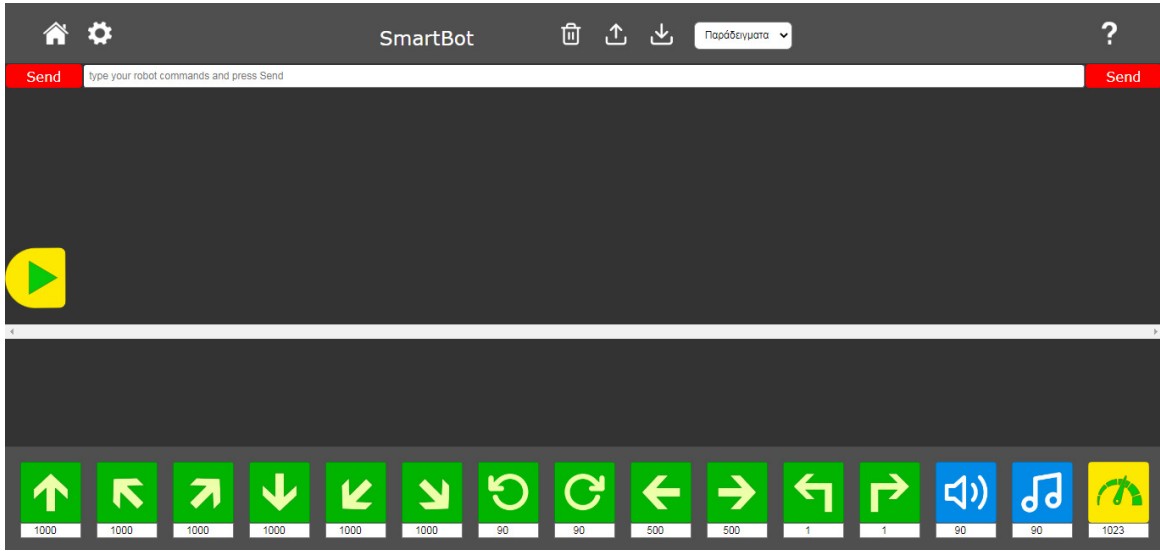

**Figure 9.** Robot's Visual Programming Language user interface and a program example.

Hard mode enriches the robot's VPL with more advanced blocks and a simulation operation where the user may evaluate the robot's program by executing it in the virtual world, but it is under development, and its final specs will be defined according to the users' feedback.

## 6. The Technology Acceptance Model

In 1989, Davis developed the Technology Acceptance Model (TAM) to predict and explain the factors that lead to the use of Information Systems [30,31]. TAM is based and is, at the same time, an adaptation of Ajzen and Fishbein's [32] Theory of Reasoned Action

(TRA) to identify those variables that are most appropriate in investigating the acceptance of the use of Information Systems. It consists of the following key factors (Figure 10) that try to explain the technology acceptance of a system; External Variables (EV), Perceived Ease of Use (PEOU), Perceived Usefulness (PU), Attitude Towards Use (ATU), and the Behavioral Intention to Use (BIU), which leads to Actual Usage (AU). User acceptance is a prerequisite for technology effectiveness [33], and this can be assessed by measuring the PEOU and PU TAM model's factors.

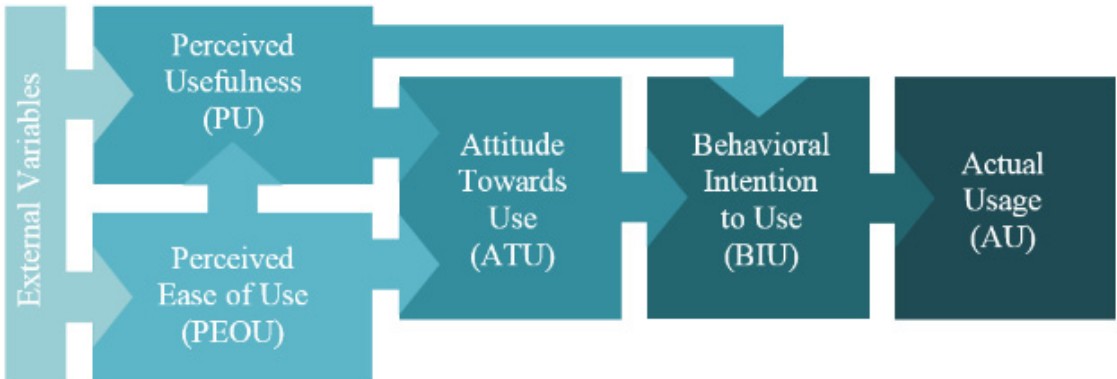

**Figure 10.** The Technology Acceptance Model (TAM).

### 6.1. External Variables

External variables are variables that shape the PEOU and PU. To name some: system's characteristics, user experience, objective design, trust, security, system features that boost the user's productivity, learning based on feedback or other individual users' factors such as age, gender, technology skills, level of education, experience, training in computer use, etc. [33–35].

### 6.2. Perceived Ease of Use

The PEOU factor refers to how a user believes the ease of using a system does not require too much effort. It is determined by External Variables (Equation (1)) and directly affects PU and ATU [30].

$$PEOU = \text{External Variables} \tag{1}$$

### 6.3. Perceived Usefulness

The PU factor is the degree to which a user believes that using the system will improve his/her performance. PU has a direct effect (Equation (2)) from PEOU and External Variable and directly affects BIU over and above ATU [30].

$$PU = PEOU + \text{External Variables} \tag{2}$$

### 6.4. Attitude towards Use

ATU refers to the evaluation of the system by the user and its attitude towards use. According to the TAM model, ATU is jointly determined by PU and PEOU (Equation (3)), with relative weights statistically estimated by linear regression [30]:

$$ATU = PU + PEOU \tag{3}$$

### 6.5. Behavioural Intention to Use

BIU factor is the intention to use and can measure the likelihood of using a system. Equation (4) shows that BIU directly affects ATU and PU [30].

$$BIU = ATU + PU \tag{4}$$

*6.6. Actual Use*

Finally, AU refers to the actual use of the system (or system usage) as measured by the duration or frequency of use of the system.

**7. Research Methodology**

*7.1. Research Context and Participants*

The next step toward the robot's development and this research's proposed AR model is the robot's evaluation by the educational community: teachers, students, and stakeholders, whose suggestions for improvements will lead to a new AR cycle [16]. Pre-service and in-service teachers were selected for the current study due to their current students or future teachers. When this research was conducted, COVID-19's restrictions limited the researchers' access to student classes, so students' evaluation was postponed for a future time.

Robots' acceptance by educators was evaluated by 116 Pedagogical and Technological Education School undergraduate and postgraduate students. A basic script of a 10-moves series was given to the students to measure the PU, PEOU, ATU, and BIU factors with a questionnaire. Students were of different gender, ages, backgrounds, studies, and levels of graduate studies (undergraduate vs postgraduate). These and others, such as the system's characteristics and features, user experience, robot design, etc., are the External Variables of the system (robot). Last but not least, prior to the research, all ethical issues [19] were taken seriously, and students filled out a consent form that informed the study's objectives and stated the procedure and the terms of the research [36].

*7.2. Instrument Development*

The research instrument consisted of two main parts: students' demographic information (gender, age, study, and teacher's speciality) and questions related to the TAM factors' construction. Four descriptive questions were added to collect qualitative data on the robot's specifications and related educational activities. An online questionnaire form was developed based on the TAM model, and empirical data were collected having 22 items.

*7.3. Demographic Statements*

This research's instrument was given to 116 participants (N = 116), undergraduate and postgraduate students—pre-service and in-service teachers—of a Greek Pedagogical and Technological Education School. Of all the responses that existed, there were no missing values (0%), same answers (0%), blank (0%), or not returned questionnaires (0%), so a 100% response rate was used in the final study. As shown in Table 2, 74 (63.8%) of the study's participants were women, and 42 (36.2%) were men, which is ordinary for a Pedagogical School in Greece [37]. The maximum response rate was gathered from participants between 22–and 34 years old (57–49.1%) and undergraduate university graduates (60–51.7%) that have a teaching speciality in science (70–70.0%). However, 16 (13.8%) of the participants were not teachers.

**Table 2.** Demographic information.

| Frequencies of Gender | | | |
|---|---|---|---|
| **Levels** | **Counts** | **% of Total** | **Cumulative %** |
| Male | 42 | 36.2 | 36.2 |
| Female | 74 | 63.8 | 100.0% |
| Frequencies of Age | | | |
| **Levels** | **Counts** | **% of Total** | **Cumulative %** |
| 22-34 | 57 | 49.1 % | 49.1 % |
| 35-44 | 43 | 37.1 % | 86.2 % |
| 45-54 | 15 | 12.9 % | 99.1 % |
| 55-64 | 1 | 0.9 % | 100.0 % |

**Table 2.** *Cont.*

| Frequencies of Study | | | |
|---|---|---|---|
| **Levels** | **Counts** | **% of Total** | **Cumulative %** |
| MSc/PhD | 55 | 47.4% | 47.4% |
| University graduates (AEI/TEI) | 60 | 51.7% | 99.1% |
| Vocational institute graduates (IEK) | 1 | 0.9% | 100.0% |
| **Frequencies of Teaching Specialty** | | | |
| **Levels** | **Counts** | **% of Total** | **Cumulative %** |
| I am not a teacher | 16 | 13.8% | 13.8% |
| Science | 70 | 70.0% | 74.1% |
| Humanitarian | 30 | 30.0% | 100.0% |

## 8. Research Framework

In this study, the TAM model was chosen to investigate the participants' (pre-service and in-service teachers) intention to use the robot for educational purposes. According to Davis [30], four instruments items are enough for the TAM's PEOU and PU construction, so the survey's PEOU and PU construct contained four items each that were modified to the context of this study, as shown in Table 3.

**Table 3.** PEOU, PU items.

| PEOU and PU Items: |
|---|
| **PEOU items:** |
| A1. It would be easy for me to learn to use the robot. |
| A2. It would be easy for me to use the robot the way I want. |
| A3. It would be easy for me to become proficient in using the robot. |
| A4. I would consider the robot easy to use. |
| **PU items:** |
| B1. Using the robot would improve my performance in my job (as a teacher). |
| B2. Using the robot in my work (as a teacher) would increase my productivity. |
| B3. Using the robot would enhance my efficiency in my work (as a teacher). |
| B4. I would find the robot useful for my work (as a teacher). |

ATU and BIU factors were measured, too, but Actual Use was not measured as the robot was not available for mass production and distribution. The original Davis' TAM items were adapted to the research's needs, translated into Greek by a professional translator, and returned to English to ensure translation equivalence [37]. All these items were measured with a 5-point Likert scale from 1 to 5, where 1—Strongly disagree, 2—Somewhat disagree, 3—Neither agree nor disagree, 4—Somewhat agree, and 5—Strongly agree. PEOU and PU were the independent variables, while ATU and BIU were the dependent variables (Table 4).

**Table 4.** Dependent and Independent Variables of the research model.

| Variable Type | Variable Codes |
|---|---|
| Independent Variables | Perceived Ease of Use (PEOU) <br> Perceived Usefulness (PU) |
| Dependent Variables | Attitude Towards Use (ATU) <br> Behavioural Intention to Use (BIU) |

## 9. Research Model and Hypotheses

The proposed research model and its hypothesized relationships between the variables that measure the participants' (teachers') intention to use the robot are shown in Figure 11. Therefore, three hypotheses based on the TAM model diagram in the context of the robot system were put forward:

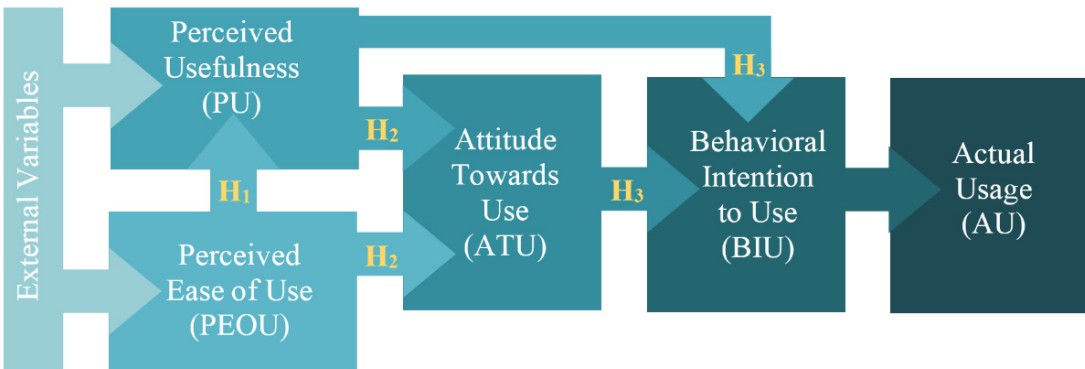

**Figure 11.** Modified TAM model for measuring the robot's intention for use.

**Hypothesis 1 (H1):** Perceived Ease of Use (PEOU) will have a significant influence on participants' Perceived Usefulness (PU) towards robot use.

**Hypothesis 2 (H2):** Perceived Ease of Use (PEOU) and Perceived Usefulness (PU will have a significant influence on participants' Attitudes Toward Use (ATU).

**Hypothesis 3 (H3):** Perceived Usefulness (PU) and Attitude Towards Use (ATU) will have a significant influence on participants' Behavioral Intention to Use (BIU) towards robot use.

In the present research, the relationships between External and PU/PEOU variables, as well as the relationship of BIU with AU, were not examined, as this research was a pilot study to obtain initial evaluation data and a primary index of intention to use, to continue to a new AR cycle for the robot's development.

## 10. Results

### 10.1. Validity and Reliability Analysis

At the outset, the reliability and validity of the model's variables (PEOU and PU construct) were first used to check. The reliability of a variable reports how precisely the measure is and refers to the repeatability or consistency of measurement. The validity reports how accurate the measure is and refers to the correctness of the measurement [38]. Cronbach Alpha ($\alpha$) coefficient and McDonald's Omega ($\omega$) coefficient were used for the reliability analysis. The popular statistic Cronbach Alpha ($\alpha$) coefficient is used to check the internal consistency, whereas McDonald's Omega ($\omega$) coefficient is an indicator of reliability in the sense of internal consistency [38,39]. For the validity analysis, Exploratory Factor Analysis (EFA) was used. EFA can be used when there is no strong a priori regarding the components of the structure intended to measure [40]. In addition, the Average Variance Extracted (AVE) for all constructs was examined to evaluate the convergent validity of the correlation between the multiple indicators of an aspect [37]. The following statistical analyses were conducted with the free, open-source, R-based software, Jamovi [41].

### 10.2. Reliability Analysis

The PEOU survey displayed an average score of 4.16 (SD = 0.67; Table 5), showing the participants' perceived ease of use. The survey was reliable, with a Chronbach's alpha of 0.84 and a McDonald's Omega ($\omega$) coefficient of 0.84. Cronbach's alpha values exceeding

0.70 are generally used as the threshold for an acceptable reliability coefficient [42,43]. The scale's reliability would not be improved by removing any survey items.

**Table 5.** Reliability Analysis for PEOU.

| | Scale Reliability Statistics | |
| --- | --- | --- |
| | Cronbach's $\alpha$ | McDonald's $\omega$ |
| scale | 0.840 | 0.841 |
| | Item Reliability Statistics | |
| | if item dropped | |
| | Cronbach's $\alpha$ | McDonald's $\omega$ |
| PEOU1 | 0.791 | 0.792 |
| PEOU2 | 0.792 | 0.797 |
| PEOU3 | 0.818 | 0.820 |
| PEOU4 | 0.787 | 0.794 |

The PU survey displayed an average score of 4.22 (SD = 0.70; Table 6), showing a strongly perceived usefulness by the participants. The survey was reliable, with a Cronbach's alpha of 0.88 and a McDonald's Omega ($\omega$) coefficient of 0.88. The scale's reliability would not be improved by removing any survey items. Furthermore, ATU and BIU surveys displayed an average score of 4.34 (SD = 0.75) and 4.05 (SD = 0.83), respectively, indicating that participants showed a strong attitude towards and intention to use the robot.

**Table 6.** Reliability Analysis for PU.

| | Scale Reliability Statistics | |
| --- | --- | --- |
| | Cronbach's $\alpha$ | McDonald's $\omega$ |
| scale | 0.878 | 0.881 |
| | Item Reliability Statistics | |
| | if item dropped | |
| | Cronbach's $\alpha$ | McDonald's $\omega$ |
| PU1 | 0.830 | 0.840 |
| PU2 | 0.853 | 0.857 |
| PU3 | 0.816 | 0.820 |
| PU4 | 0.872 | 0.873 |

*10.3. Validity Analysis*

In the beginning, two assumptions (sphericity and sampling adequacy) had to be checked as part of the EFA. Bartlett's test for sphericity was used to check whether the observed correlation matrix diverges significantly from a null correlation matrix. It was found (Table 7) that Barlett's test *p*-value < 0.001 (*p* must be less than 0.05 for test significance), meaning the test was significant and the first assumption was satisfied [38]. Kaiser-Meyer-Olkin (KMO) Measure of Sampling Adequacy (MSA) was used to check sampling adequacy. The KMO index measures the proportion of Variance among observed variables that might be common; the higher ($\approx$1) the KMO index variance, the more relevant EFA is. The overall KMO measure of MSA was found equal to 0.78 (Table 7), meaning good sampling adequacy [38], so the second assumption was satisfied.

**Table 7.** EFA's Assumption Checks.

| Bartlett's Test of Sphericity | | |
|:---:|:---:|:---:|
| $\chi^2$ | df | $p$ |
| 472 | 28 | < 0.001 |
| KMO Measure of Sampling Adequacy | | |
| | MSA | |
| Overall | 0.777 | |
| PU1 | 0.784 | |
| PU2 | 0.699 | |
| PU3 | 0.789 | |
| PU4 | 0.870 | |
| PEOU1 | 0.763 | |
| PEOU2 | 0.826 | |
| PEOU3 | 0.700 | |
| PEOU4 | 0.790 | |

Next, the Eigenvalues of Table 8 were checked to find the factors to be used with the research's data. Eigenvalues more significant than 1 indicate a factor, so two factors were found suitable for this research's data [38].

**Table 8.** EFA's Initial Eigenvalues.

| Factor | Eigenvalue |
|:---:|:---:|
| 1 | 3.2944 |
| 2 | 1.2089 |
| 3 | −0.0356 |
| 4 | −0.0673 |
| 5 | −0.1307 |
| 6 | −0.2141 |
| 7 | −0.3545 |
| 8 | −0.4065 |

Table 9 shows the EFA's factor loadings onto each selected factor. Both factors and the factor loadings match the putative factors specified in the research's model. Uniqueness is defined as the proportion of Variance and is not explained by the factors. The lower the uniqueness, the greater the relevance or contribution of the variable in the factor model [38]. The table's nine variables' uniqueness varies between 0.21 and 0.53, which is extremely good and good.

In addition, Confirmatory Factor Analysis (CFA) was used to measure AVE, ranging from 0.57 for PEOU and 0.65 for PU, indicating convergent validity as it was greater than 0.5 for each construct [37]. Table 10 shows the uniqueness of each variable.

**Table 9.** EFA's Factor Loadings.

| | Factor | | |
| --- | --- | --- | --- |
| | **1** | **2** | **Uniqueness** |
| PU1 | 0.837 | | 0.286 |
| PU2 | 0.801 | | 0.402 |
| PU3 | 0.889 | | 0.208 |
| PU4 | 0.664 | | 0.479 |
| PEOU1 | | 0.792 | 0.392 |
| PEOU2 | | 0.780 | 0.409 |
| PEOU3 | | 0.698 | 0.528 |
| PEOU4 | | 0.740 | 0.372 |

Note. The 'Minimum residual' extraction method was combined with an 'oblimin' rotation.

**Table 10.** EFA's Factor Statistics.

| Factor | SS Loadings | % of Variance | Cumulative % |
| --- | --- | --- | --- |
| 1 | 2.61 | 32.6 | 32.6 |
| 2 | 2.31 | 28.9 | 61.5 |
| **Inter-Factor Correlations** | | | |
| | **1** | **2** | |
| 1 | — | 0.397 | |
| 2 | | — | |

## 11. Robot's Level of Acceptance

Firstly, Spearman's rank-order correlation evaluated all research hypothesis associations between variables to measure the participants' intention to use and understand the robot's acceptance. Spearman's correlation calculates a coefficient, *rho* (or $\rho$), which measures the strength and direction of the association/relationship between two continuous or ordinal variables. It has only three assumptions to consider [44]:

- there are two continuous or ordinal variables.
- these two variables stand for paired observations, and
- there is a monotonic relationship between the two variables.

Next, the first hypothesis (H1) based on the TAM model was evaluated using a linear regression analysis statistical test, and the other hypotheses (H2 and H3) were evaluated using multiple regression analysis. Linear regression analysis demands the following seven assumptions to be considered [45]:

- one dependent variable that is measured at the continuous level.
- one independent variable that is measured at the continuous level.
- there should be a linear relationship between dependent and independent variables.
- there should be independent observations.
- there should be no significant outliers.
- the variances along the line of best fit remain similar as you move along the line, known as homoscedasticity; and
- the residuals (errors) of the regression line are normally distributed.

**Hypothesis 1:** Perceived Ease of Use (PEOU) will significantly influence participants' Perceived Usefulness (PU) towards robot use.

First, a Spearman's correlation was run to assess the relationship between PEOU and PU. Preliminary analysis showed the relationship to be monotonic, as assessed by visual

inspection of a scatterplot. There was a statistically significant, positive correlation between PEOU and PU; *rho* = 0.31, *p* < 0.001, so the next step was to predict PU from the PEOU using linear regression analysis. All the above linear regression analysis assumptions were examined and confirmed that they were met. Both dependent and independent variables were measured at the continuous level. The visual inspection of a scatterplot (PEOU against PU) indicated a linear relationship between the variables. The independence of residuals was assessed by a Durbin–Watson statistic of 1.64. There were no significant outliers. There was homoscedasticity, as assessed by visual inspection of a plot of standardized residuals versus standardized predicted values, and the residuals were normally distributed as assessed by visual inspection of a normal probability plot. It was found the PU use was significantly predictive of PEOU, R = 0.36, F(1, 114) = 17.3, *p* < 0.001 (Table 11).

**Table 11.** Linear Regression PU = PEOU.

| Model Fit Measures | | | | | | |
|---|---|---|---|---|---|---|
| | | | | | Overall Model Test | |
| Model | R | R$^2$ | F | df1 | df2 | *p* |
| 1 | 0.363 | 0.132 | 17.3 | 1 | 114 | < 0.001 |
| Model Coefficients—Perceived Usefulness (PU) | | | | | | |
| Predictor | Estimate | SE | t | | *p* | |
| Intercept | 2.634 | 0.3852 | 6.84 | | <0.001 | |
| Perceived Ease of Use (PEOU) | 0.380 | 0.0915 | 4.16 | | <0.001 | |

**Hypothesis 2:** Perceived Ease of Use (PEOU) and Perceived Usefulness (PU will have a significant influence on participants' Attitudes Toward Use (ATU).

For H2 and H3 evaluation, the following multiple linear regression analysis assumptions had to be considered first [46]:

- one dependent variable that is measured at the continuous level.
- two or more independent variables that are measured at the continuous level.
- independence of observations (i.e., independence of residuals).
- there should be a linear relationship between dependent and the (each of, and collectively) independent variables.
- there should be homoscedasticity of residuals (equal error variances).
- data must not show multicollinearity (two or more independent variables that are highly correlated with each other).
- there should be no significant outliers, highly influential points, or high leverage points; and
- the residuals (errors) should be normally distributed.

First, a Spearman's correlation was run to assess the relationship between PEOU, PU, and ATU. Preliminary analysis showed that the relationships meet the above Spearman's correlation associations. There was a statistically significant, strong positive correlation between PEOU and ATU, *rho* = 0.79, *p* < 0.001, and there was a statistically significant, strong positive correlation between PU and ATU, *rho* = 0.81, *p* < 0.001, so the next step was to try to predict ATU from PU and PEOU using multiple linear regression analysis. After the multiple linear regression analysis assumptions were examined, it was found that the ATU use was significantly predictive (correlated) of jointly determined by PEOU and PU, R = 0.74, F(2, 113) = 68.5, *p* < 0.001 (Table 12).

**Table 12.** Multiple Linear Regression ATU = PU + PEOU.

| Model Fit Measures | | | | | | | |
|---|---|---|---|---|---|---|---|
| | | | | Overall Model Test | | | |
| Model | R | $R^2$ | Adjusted $R^2$ | F | df1 | df2 | *p* |
| 1 | 0.740 | 0.548 | 0.540 | 68.5 | 2 | 113 | < 0.001 |
| Model Coefficients—Attitude Toward Use (ATU) | | | | | | | |
| Predictor | | | | Estimate | SE | t | *p* |
| Intercept | | | | 1.0585 | 0.3549 | 2.982 | 0.004 |
| Perceived Ease of Use (PEOU) | | | | −0.0204 | 0.0762 | −0.268 | 0.789 |
| Perceived Usefulness (PU) | | | | 0.7997 | 0.0727 | 11.005 | < 0.001 |

**Hypothesis 3:** Perceived Usefulness (PU) and Attitude Towards Use (ATU) will have a significant influence on participants' Behavioral Intention to Use (BIU) towards robot use.

Spearman's correlation was run to assess the relationship between PU, ATU, and BIU. Preliminary analysis showed that the relationships meet the above Spearman's correlation associations. There was a statistically significant, strong positive correlation between PU and BIU, *rho* = 0.60, *p* < 0.001, and there was a statistically significant, strong positive correlation between ATU and BIU, *rho* = 0.65, *p* < 0.001, so the next step was to try to predict BIU from PU and ATU using multiple linear regression analysis. As in case H2, in the case of H3, the multiple linear regression analysis assumptions were examined. It was found the BIU use was significantly predictive (correlated) of jointly determined by ATU and PU, R = 0.66, F(2, 113) = 43.1, *p* < 0.001 (Table 13).

**Table 13.** Linear Regression BIU = ATU + PU.

| Model Fit Measures | | | | | | | |
|---|---|---|---|---|---|---|---|
| | | | | Overall Model Test | | | |
| Model | R | $R^2$ | Adjusted $R^2$ | F | df1 | df2 | *p* |
| 1 | 0.658 | 0.433 | 0.423 | 43.1 | 2 | 113 | < 0.001 |
| Model Coefficients—Behavioral Intention to Use (BIU) | | | | | | | |
| **Predictor** | | | | **Estimate** | **SE** | **t** | *p* |
| Intercept | | | | 0.634 | 0.380 | 1.67 | 0.098 |
| Perceived Usefulness (PU) | | | | 0.255 | 0.126 | 2.03 | 0.045 |
| Attitude Toward Use (ATU) | | | | 0.539 | 0.117 | 4.59 | < 0.001 |

## 12. Discussion

The results show that most participants had a prominent level of acceptance for the robot, which is quite promising for the robot's use and further research and development. The participants tend to (Table 14): (i) strongly agree that the robot is ease of use $\bar{x}$ = 4.16 (out of a maximum of 5), SD = 0.67, (ii) strongly believe that using the robot will improve his/her performance $\bar{x}$ = 4.22, SD = 0.70, (iii) strongly agree towards attitude to robot use $\bar{x}$ = 4.34, SD = 0.75, and (iv) strongly agree for their intention to use the robot $\bar{x}$ = 4.05, SD = 0.83.

Furthermore, the TAM model's hypothesis (H1, H2, and H3) was evaluated and confirmed, showing a significant correlation between (i) PU and PEOU use, (ii) ATU and jointly determined PEOU, PU, and (iii) BIU and jointly determined ATU, PU (Figure 12).

**Table 14.** Descriptive statistics for variables.

|  | PEOU | PU | ATU | BIU |
|---|---|---|---|---|
| N | 116 | 116 | 116 | 116 |
| Missing | 0 | 0 | 0 | 0 |
| Mean | 4.16 | 4.22 | 4.34 | 4.05 |
| Standard deviation | 0.666 | 0.698 | 0.747 | 0.832 |

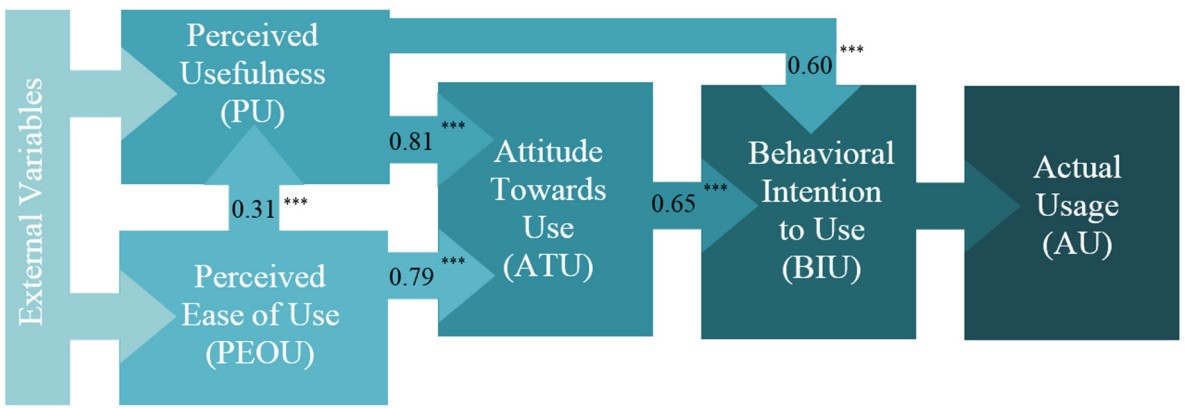

**Figure 12.** Results of the proposed research model. Note: * $p < 0.05$, ** $p < 0.01$, *** $p < 0.001$.

When this research was conducted, the researchers' access was limited to student classes due to COVID-19 restrictions, so applying the proposed robot in the educational process was impossible. However, shortly, the researchers will proceed to the pedagogical utilization of the educational robot, and in the following paper, extensive results will be presented.

An initial future design includes the application of the robot in educational activities to investigate the possibility of developing computational thinking skills in primary school students by applying educational robotics activities when they are asked to solve authentic STEM problems. Specifically, the computational thinking skills of abstraction, generalization, algorithm, modularity, segmentation, debugging, and collaboration are intended to be researched.

## 13. Conclusions

In this research, an educational robot inspired by the results of a survey and developed by AR is evaluated by the TAM model with a positive impact which is particularly encouraging for its development. The user's acceptance of technology indicates a positive psychological status towards the usage intention ensuring the robot's success. Moreover, technology acceptance must constantly follow users' requirements towards the technology life cycle [47]. Other researchers [48–52] have also used the TAM model to evaluate their robots and had similar positive results, or even extended the TAM model to their robot's special needs [53].

However, this study has some limitations. First, COVID-19's restrictions limited the researchers' access to classes, so students' evaluation was absent. Second, only the TAM model was used for evaluation. It would be more reliable if other models, e.g., Unified Theory of Acceptance and Use of Technology (UTAUT), Innovation Diffusion Theory (IDT), Website Analysis and Measurement Inventory (WAMMI), TAM3, etc., could be used in conjunction with this research. Third, external variables, e.g., participant-related factors, were not considered. Previous studies have indicated that factors such as gender and age can affect users' perspectives [52]. Hence, a future study should concern with (i) including students, teachers, parents, and other stakeholders in the research's participants, (ii) collecting more variables data based on the different evaluation models, and qualitative

data such as field observation, interviews, recordings, videotaping, etc., and (iii) examine the relationships between external variables such gender, age, technology skills, and level of education with their intention to use the robot.

**Author Contributions:** Data curation, A.C.; Formal analysis, A.C. and M.P.; Investigation, A.C. and M.K.; Methodology, M.P.; Project administration, M.K.; Resources, S.P.; Visualization, S.P.; Writing—original draft, S.P. and M.P.; Writing—review & editing, S.P. All authors have read and agreed to the published version of the manuscript.

**Funding:** This research received no external funding.

**Institutional Review Board Statement:** The study was conducted following the Declaration of Helsinki. No ethical approval is required.

**Informed Consent Statement:** Informed consent was obtained from all subjects involved in the study.

**Data Availability Statement:** The data that support the findings of this study are available from the corresponding author upon reasonable request.

**Conflicts of Interest:** The authors declare no conflict of interest.

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
