# Peer review of "A Novel, Modular Robot for Educational Robotics Developed Using Action Research Evaluated on Technology Acceptance Model"

_education, doi:10.3390/educsci12040274_

Round 1
Reviewer 1 Report
The manuscript is interesting and well-structured. It presents clearly the results achieved with the research implemented, including appropriate figures and tables.
However, the suggestion is that the discussion on the connection between the factor "A" (in STEAM) with the current educational platforms should be more detailed than done in paragraphs Par.73; Par. 139-141; Par 192.
As described, it seems the shells' design of the robots can be enough to transform STEM into STEAM. The factor “A” is something in more including non-only “Arts” (in the strict sense, e.g. design) but also humanities science. This means that what engages students in “A” factor for an educational robotics activity should not be limited to designing different shells. It depends on how the teachers can integrate and combine the STEAM topics to engage students in a multidisciplinary manner. This can be reached mainly through specific pedagogical approaches and preparing appropriate lesson plans. Otherwise, it seems to be too reductive.
Actually, STE"A"M topics are not sufficiently discussed in the manuscript. In fact, although the cited references are mostly current, there is no relevant bibliography referring to the factor “A” in STEM.
Author Response
Dear Reviewer 1,
We were very pleased to receive the your comments on our manuscript (education-1671210) with the initial title “A novel, modular robot for Educational Robotics; developed using Action Research evaluated on Technology Acceptance Model”.
We are sincerely grateful for your help and comments and suggestions that made our contribution richer.
When preparing our revised manuscript, we have promptly and carefully considered all the suggestions. We have responded to the requested revisions, and the changes have been made also within the revised manuscript.
The point-by-point answers to the comments are mentioned with track changes within the revised text.
Thank you again for your help and your consideration.
Kind regards
Authors
Reviewer 2 Report
The article presents a well written, theoretically underlined and well conducted investegation of the use of modular robots in education. The work is linked to existing litterature. However, some of the methological choises seems a little odd, which will be the focus of my following comments.
Keep in mind that this is a minor flaw, as the overall robot acceptance level is calculated with the correct spearman correlation for ordinal data. However, I wonder why:
- Table 5 presents mean and standard deviation of what seems to be the ordinal questioneer data. This add little value, while might be a potential flaw (Lindell)
- The use of Cronbachs alpha can be justified, but an ordinal alpha would be a more safe bet. (Gadermann)
- The same applies to McDonalds Omega, compared to an ordinal omega meassure (Gadermann).
Lastly, the discussion is rather short for such an ells comprehensive article. What are the implications of the results towards learning, and how does it align with other relevant bodies of theory.
Liddell, Torrin M., and John K. Kruschke. "Analyzing ordinal data with metric models: What could possibly go wrong?." Journal of Experimental Social Psychology 79 (2018): 328-348.
Gadermann, Anne M., Martin Guhn, and Bruno D. Zumbo. "Estimating ordinal reliability for Likert-type and ordinal item response data: A conceptual, empirical, and practical guide." Practical Assessment, Research, and Evaluation 17.1 (2012): 3.
Author Response
Dear Reviewer 2,
We were very pleased to receive the your comments on our manuscript (education-1671210) with the initial title “A novel, modular robot for Educational Robotics; developed using Action Research evaluated on Technology Acceptance Model”.
We are sincerely grateful for your help and comments and suggestions that made our contribution richer.
When preparing our revised manuscript, we have promptly and carefully considered all the suggestions. We have responded to the requested revisions, and the changes have been made also within the revised manuscript.
The point-by-point answers to the comments are mentioned with track changes within the revised text.
Thank you again for your help and your consideration.
Kind regards
Authors